# A Novel Self-Attention Mechanism-Based Dynamic Ensemble Model for Soil Hyperspectral Prediction

**DOI:** 10.3390/s26010195

**Published:** 2025-12-27

**Authors:** Keyang Yin, Jia Deng, Huixia Li, Chunhui Feng, Jie Peng

**Affiliations:** 1College of Agriculture, Tarim University, Alar 843300, China; 2College of Horticulture and Forestry, Tarim University, Alar 843300, China; 3Key Laboratory of Genetic Improvement and Efficient Production for Specialty Crops in Arid Southern Xinjiang of Xinjiang Corps, Alar 843300, China

**Keywords:** ensemble method, visible–near-infrared spectroscopy, dynamic weight allocation, soil organic matter, data structure

## Abstract

**Highlights:**

**What are the main findings?**
Dynamic weight assignment is effective for weighted averaging ensemble models.Weighting methods based on training process information outperform traditional evaluation-index-based methods.The self-attention mechanism provides the most effective weight allocation.The best ensemble performance is achieved with 26 base learners.

**What is the implication of the main finding?**
Dynamic weight allocation enhances the generalization performance and robustness of ensemble models and reduces sensitivity to outliers and noise.This study provides scientific theoretical support for high-accuracy SOM monitoring using Vis–NIR spectroscopy.

**Abstract:**

Visible–near-infrared spectroscopy enables rapid, non-destructive soil organic matter (SOM) detection, yet its prediction accuracy relies heavily on the effectiveness of the chosen algorithmic models. Weighted Averaging Ensemble Models (WAEM) are robust but face a key challenge: their performance depends on optimal base learner weight allocation, which standard evaluation indices often fail to achieve, risking biased weights and local optima. This study significantly enhances WAEM by determining optimal weights using information extracted from the model training process via seven methods, including reinforcement learning and a self-attention mechanism (Sam). Experiments on 704 soil samples from China’s Tarim River Basin employed a dynamic data structure for real-time weight updating. Results show that six WAEM methods utilizing training process information outperformed conventional evaluation index approaches. Improvements reduced WAEM root mean square error (RMSE) by 0.028–1.279 g kg^−1^ and increased the correlation coefficient (R^2^) by up to 0.06. Sam achieved the highest performance, with R^2^ and RMSE reaching 0.927 and 2.325 g kg^−1^, respectively. Furthermore, model R^2^ began converging at 26 base learners, indicating diminishing returns from adding more. This research confirms that dynamic WAEM weight allocation via Sam significantly boosts SOM prediction accuracy, providing a scientific foundation for infrared-based soil monitoring.

## 1. Introduction

According to the Food and Agriculture Organization’s (FAO) Global Food Security and Nutrition Report 2022, over 700 million individuals worldwide are grappling with hunger. The report further underscores the critical necessity to enhance soil health and boost agricultural productivity [1]. Soil organic matter (SOM) is a crucial element in assessing the sustainability of soil health and productivity. It significantly contributes to optimizing soil structure, preserving water content, supplying essential nutrients to plants, and enhancing microbial activity [2,3,4,5]. Therefore, the rapid and precise inversion of SOM content holds immense significance for gaining insight into soil quality and productivity, assessing soil fertility, and achieving the sustainable development of farmland [6]. However, traditional methods for testing SOM necessitate intricate laboratory procedures and time-consuming processes for sample preparation and handling [7,8]. In recent years, the visible–near-infrared (Vis-NIR) spectroscopy technique has gained widespread application as a swift and non-invasive method for detecting SOM content [9,10].

Numerous studies have validated the efficacy of combining Vis-NIR spectroscopy with Machine Learning (ML) techniques for the effective monitoring of SOM content [9,11,12]. However, traditional ML models frequently suffer from the drawback of inadequate performance when applied to unfamiliar or unseen datasets [13]. To address this issue, certain studies have attempted to employ EM characterized by high generalization performance and robust resilience, integrating them with Vis-NIR spectroscopy [14,15].

The advantage of the integrated model (EM) lies in its unique architecture, which can more effectively capture the highly complex and nonlinear mapping relationship between soil spectra and SOM. This relationship is rooted in the multi-scale heterogeneity of the soil itself [16]. From a soil science perspective, SOM is not a homogeneous entity but rather a continuum composed of plant residues at different decomposition stages, microbial metabolites, and stable humus. Its chemical structure (such as alkyl, aromatic, and carboxyl content) and physical state (such as the way it combines with minerals) show significant differences [10,17]. These differences result in subtle but crucial characteristic responses in the Vis-NIR spectrum, such as shifts in specific absorption peaks, changes in the intensity of broad absorption bands, and differences in multiple scattering effects. Traditional single machine learning models or even some deep learning models may have difficulty simultaneously and evenly capturing the underlying linear trends (such as the negative correlation between overall organic carbon content and broad-band reflectance) and deep nonlinear patterns (such as the complex interaction between specific organic components and narrow-band absorption features) when faced with this spectral-attribute relationship formed by multiple biogeochemical processes coupling [17]. The integrated model, through strategic combination of multiple base learners, can extract diverse feature interactions from different subspaces or data samples [16]. This mechanism enables it to more precisely analyze the mixed and overlapping signals in the spectrum, thereby simultaneously modeling the combined effects of humus stabilization (possibly manifested as gradual changes in specific spectral bands) and fresh organic matter input (possibly triggering local spectral mutations) on the spectrum, ultimately establishing a more robust and interpretable predictive relationship that effectively enhances the generalization ability for unknown or heterogeneous environmental samples.

In the domain of EM, the weighted averaging (WA) EM, renowned for its robust capabilities, has been extensively utilized across diverse classification and regression tasks due to its strong generalization properties and high level of explainability [18,19]. At the heart of WA lies its efficient amalgamation of predictions from multiple base learners via a meticulously designed weight allocation mechanism. These weights not only precisely capture the contribution of each base learner towards the ultimate outcome but also adeptly capitalize on their individual strengths to rectify biases like overestimation at low values or underestimation at high values, consequently bolstering the robustness of EM [20,21]. Hence, the reasonableness of the weight allocation among base learners is a pivotal determinant in ascertaining the predictive performance of the WA within EM.

However, within the current research landscape of ensemble learning, a significant portion of studies predominantly hinges on the performance indices of the base learners to compute the weights [22,23,24]. This method, although intuitive and straightforward to execute, carries substantial limitations. Primarily, the accuracy of weight distribution largely relies on the efficacy of the chosen evaluation indicators. Should these metrics fail to fully capture the true capabilities of the learner, the resultant weight assignments could stray from the ideal values, thereby compromising the overall efficacy of the EM. For instance, certain criteria might excel on specific datasets yet falter on others. Additionally, an overemphasis on evaluation indices to dictate weight allocations may overlook the distinctions and diversity among various base learners. A key advantage of ensemble learning is its capacity to amalgamate the strengths of multiple models, with diversity being a critical component in attaining this synergy [21]. Simultaneously, if the allocation of weights is determined solely by the model’s performance metric, it could result in the undervaluation of certain base learners that possess unique strengths yet perform poorly on that specific evaluation metric. This oversight can diminish the overall potential performance of the EM. Furthermore, in real-world applications, data frequently contain noise or outliers, which can significantly influence the evaluation indicators. Consequently, this may skew the weight distribution among some base learners, impacting the model’s stability and its ability to generalize to new data [20].

In recent years, advancements in artificial intelligence (AI) technology have introduced a suite of sophisticated methodologies including the self-attention mechanism (Sam), reinforcement learning (RL), and adaptive learning methods (AL) [25,26]. These innovations offer a fresh approach to weight computation, allowing us to capture complex relationships within data hierarchies during the learning process. As a result, these techniques have been embraced across various domains, enabling algorithms to adeptly accommodate diverse data distributions and patterns [27,28,29]. Leveraging the capability of these methods to discern the interplay between trajectory and data in model learning, we formulate a hypothesis suggesting that their application to the computation of base learner weights could yield more rational and precise weight allocations. This proposed approach would rely on the insights gleaned from the training process records of the base learner, potentially surpassing traditional methods that depend on base learner evaluation metrics alone.

Another critical consideration is the limitation of weights derived from static evaluation indicators in adapting to shifts in data distribution. In numerous real-world contexts, data distributions can evolve dynamically over time. This necessitates that the weights within the EM are capable of dynamic adjustment, thereby enabling the model to flexibly adapt to the evolving data landscape [30]. Nonetheless, the challenge lies in the static weight allocation strategy frequently falling short in fulfilling this requirement, thereby constraining the efficacy of the EM in dynamic environments. Addressing this issue, the linked list emerges as a viable solution. As a linear data structure, it facilitates the dynamic storage and management of elements by leveraging node-to-node pointer links. This capability offers a potential avenue for achieving dynamic weight allocation, enhancing the adaptability of EM in fluctuating data scenarios [31]. We hypothesize that the adaptable node management and update mechanisms intrinsic to linked lists can more effectively accommodate the dynamic shifts in data distribution. Thereby enabling timely adjustments to the weights assigned to each base learner. Moreover, the selection of the base learner count is a critical issue that merits attention. An excessive number of base learners could lead to model overfitting, whereas an insufficient quantity might hinder the full potential of the EM [22,23]. Current research has yet to identify the optimal number of base learners for the WA EM. Consequently, ascertaining the ideal count of base learners for the WA EM is an imperative challenge that requires immediate attention.

To summarize, the existing WA EM faces limitations in weight allocation, chiefly characterized by its dependence on evaluation indicators, ignorance of model diversity, sensitivity to outliers, and an incapacity to adapt dynamically to shifting data trends. Additionally, the ambiguity surrounding the determination of the optimal number of base learners within the WA EM framework significantly constrains the performance of the ensemble method. To address these challenges, the study utilized the SOM content of 704 soil samples (depth 0–20 cm) from the Tarim River basin in Xinjiang, China, as the research indicator. We innovated the weight allocation strategy, constructing a WA EM with 34 base learners, utilizing seven distinct weight allocation methods to extract information from the process. A linked list data structure was developed to facilitate dynamic weight allocation, and the selection of the base learner count within the WA EM was investigated. The objective was to assess the prediction efficacy of the WA EM employing these methodologies on a substantial dataset of soil samples, identifying the superior weight allocation approach and optimal number of base learners to enhance the precision and reliability of SOM content predictions by WAEM. The specific objectives encompass: (1) from the perspective of the model-based training process, a novel method for calculating the weights of base learners is proposed; (2) Develop a dynamic method for allocating the weights of base learners to achieve real-time and adaptive adjustment of the contribution of each model, thereby maximizing the prediction performance and stability of WAEM; (3) determining the optimal base learner count for WAEM; (4) Construct a highly accurate SOM-based inverse WAEM.

## 2. Materials and Methods

### 2.1. Study Area and Soil Sample Collection

The Tarim River basin, situated in the southern part of the Xinjiang Uygur Autonomous Region, has been designated as the research zone. A total of 704 soil samples were systematically gathered from six diverse regions, namely Xinhe County (Figure 1A), Baicheng County (Figure 1B), Shaya County (Figure 1C), Hetian County (Figure 1D), Awati County (Figure 1E), Aral County (Figure 1F) and Wensu county (Figure 1G), with depths ranging from 0 to 20 cm. Each sampling site was positioned at the heart of the field and situated at least 50 m away from any boundary. This region was chosen for the study due to its representation and significant variation in soil data, influenced by factors such as geomorphology, parent material, soil type, and climatic characteristics [12]. To accurately map the sampling locations, we utilized a handheld Global Positioning system (GPS, Trimble AG25, Sunnyvale, CA, USA), ensuring the recording of precise geographical coordinates for each site. The study area’s precise boundaries and the distribution of sampling points are vividly illustrated in Figure 1.

The Tarim River Basin holds the distinction of being the largest inland water system in China, stretching across the vast Tarim Basin that lies between the majestic Tianshan, Kunlun, and Altun mountain ranges. It spans an impressive area of 1.02 × 10^6^ km^2^. The Tarim River, measuring 2179 km in length, is formed by the confluence of several significant tributaries, including the Yerqiang River, Hetian River, Aksu River, and Weigan River. This region serves not only as a vital hub for economic and ecological preservation in southern Xinjiang but also stands as a quintessential example of arid ecosystems within China. The research area is strategically situated between the Tianshan Mountains and the Kunlun Mountains, encompassing desert and oasis landscapes and characterized by a typical temperate continental arid climate [17,32]. Within this area, the primary crops cultivated consist of cotton and red dates, with cotton alone accounting for over 80% of the total arable land. The main irrigation method employed in the agricultural planting areas is still surface flooding. Owing to the region’s scant annual rainfall (ranging from 46.4 to 64.5 mm) and substantially higher evaporation rates compared to its rainfall (which varies from 1992 to 2863.4 mm), the soil moisture content is consistently low. This results in diminished soil fertility, as evidenced by lower levels of SOM, potassium (K), and phosphorus (P). Thus, the swift assessment of SOM content within the study area holds paramount importance for improving agricultural productivity in this particular region.

### 2.2. Laboratory Analysis and Spectra Measurement of Soil Samples

During the experiment of testing the soil organic matter (SOM) content in soil samples, the first stage involves removing impurities. During this stage, clean forceps are used to manually remove obvious plant roots, stones, debris, and any non-soil inorganic fragments. After impurity removal, the samples enter the natural weathering and drying stage. To avoid the thermal decomposition effect caused by high-temperature drying (especially for light organic components), the room temperature natural air-drying method is adopted. Subsequently, the samples are evenly spread on a porcelain plate lined with clean sulfuric acid paper and placed in a ventilated, cool, and dust-free laboratory. They are turned over regularly to ensure uniform drying. This process lasts approximately 5–7 days until the samples reach constant weight (the difference between two consecutive weighings is less than 0.1%). Natural air-drying not only removes moisture but also partially disintegrates the soil aggregates, facilitating subsequent grinding. The dried samples are finely ground using a planetary ball mill. The grinding process begins by initially breaking the large soil samples into pieces, placing them in a zirconium oxide grinding jar, and grinding at 300 rpm for 15 min. Subsequently, the ground soil is passed through 2 mm and 0.2 mm (200 mesh) stainless steel standard analysis sieves [12]. The sieving process uses a mechanical vibrating screen to ensure consistent screening efficiency. The soil passing through the 2 mm sieve is used for SOM chemical analysis to meet the particle size requirements of the standard method; while the extremely fine powder passing through the 0.2 mm sieve is specifically used for vis-NIR spectroscopy measurement because more uniform and finer particle sizes can significantly reduce spectral scattering effects and yield clearer and more reproducible reflectance spectra.

The SOM content was determined using the classic external heating-chromic acid potassium volumetric method, which is based on the authoritative procedure established by Mebius et al. [33]. In the experimental operation, 0.1000 g (accurate to 0.0001 g) of soil samples sieved through a 2 mm sieve were accurately weighed in a hard test tube. 5.00 milliliters of 0.8 mol/L K_2_Cr_2_O_7_ standard solution and 5 milliliters of concentrated sulfuric acid were added, and they were thoroughly mixed before being placed in a preheated oil bath pot and heated to 175 ± 5 °C for 5 min. After cooling, the reaction solution was transferred to a conical flask, and the test tube was rinsed with deionized water. 3–4 drops of o-phenanthroline indicator were added, and the solution was titrated with 0.2 mol/L FeSO_4_ standard solution. The color change from orange-yellow to blue-green to brick red indicates the end point [32]. Blank tests (without soil samples) and standard substances (soil samples with known SOM content) were set up for quality control. All samples were subjected to double parallel determinations, with a relative deviation requirement of less than 2%. The final SOM content determination results of all soil samples in this study ranged from 0.09 g kg^−1^ to 66.46 g kg^−1^, with an average value of 14.85 g kg^−1^. The data analysis showed that the higher SOM content was significantly concentrated in paddy soil samples.

Spectral data collection was completed using the ASD FieldSpec4 high-performance surface spectral instrument. During the operation, approximately 15 g of soil powder sieved through a 0.2 mm sieve were evenly filled into a 10 cm diameter, 1.5 cm deep black circular Petri dish. The dish was scraped flat and slightly compacted to form a smooth, uniform, and non-glossy measurement surface. The measurement was conducted using a contact probe, which contained a halogen lamp as the light source. The probe was placed vertically against the soil surface to ensure no interference from ambient light. For each soil sample, 10 different points were randomly selected on the Petri dish surface for measurement, and each measurement was considered a repetition. The measurement result of each point was the average value of multiple internal scans by the instrument. Finally, the spectral reflectance of this sample was the arithmetic mean spectrum of the 10 repeated readings. This highly repetitive measurement strategy effectively reduces the measurement random errors caused by the microscopic heterogeneity of the soil.

After obtaining the spectral data of the original soil samples, the invalid bands will be removed first. Due to the decreased sensitivity of the instrument detector at the ends of the spectral range and the influence of atmospheric absorption, the signal-to-noise ratio in the 350–400 nm band is extremely low and unstable, while the 2400–2500 nm band is interfered by strong water vapor absorption. Therefore, we decisively removed the data from these two intervals from the subsequent analysis and only retained the 400–2400 nm core spectral interval with high signal-to-noise ratio and rich information for analysis. Subsequently, the Savitzky–Golay (SG) convolution smoothing algorithm was applied to denoise the spectral data. This method was recommended by Dotto et al. and is widely used in soil spectral analysis. The essence of SG smoothing is a weighted moving average method based on local polynomial least squares fitting. Its advantage is that it can effectively smooth random noise while maximizing the retention of the true characteristics of the original spectral curve (such as the width and shape of absorption peaks), avoiding the distortion of the spectrum like simple moving average. The SG smoothing parameters are: window size (Window Size) is 11 data points (i.e., 11 nm), and the polynomial order (Polynomial Order) is 2 [34].

### 2.3. Data Preprocessing and Division

Prior to constructing the predictive model, it is crucial to ensure that the calibration and validation samples are representative. To achieve this, the dataset was split into calibration and validation sets using the Kennard-Stone (K-S) algorithm, following a 2:1 ratio [35]. This division resulted in 70% of the 470 calibration samples and 30% of the 234 validation points. The average SOM content in the calibration set was found to be 15.83 g kg^−1^, with a coefficient of variation of 44.37%, while the SOM content in the validation set was slightly higher at 16.08 g kg^−1^, with a coefficient of variation of 48.66%. The data indicates that there is minimal difference in SOM content between the two sets, ensuring a balanced distribution that is suitable for developing a predictive model. Each learner undergoes training within the calibration set via 10-fold cross-validation. The performance of the model is subsequently evaluated using the validation set [35,36]. The principle of the K-S algorithm for partitioning the data set is to construct a multi-dimensional feature space composed of all sample spectral data and calculate its Euclidean distance matrix [35]. Firstly, K-S will achieve the maximum uniform coverage of the feature space through iterative selection. Then, it selects the sample with the farthest distance and closest to the overall spectral mean as the initial calibration point; subsequently, in each round of iteration, it calculates the minimum distance between each remaining sample and the current calibration set, and selects the sample with the maximum minimum distance (i.e., the point located in the “blank” area covered by the current calibration set) into the calibration set [36]. This process continues until the number of calibration set samples reaches the preset target (such as 470 in this study). Finally, the actively selected samples form a representative calibration set, while the remaining samples naturally become the validation set. This deterministic method ensures that the calibration set can widely cover the entire variation range of the sample distribution, allowing the validation set to be uniformly filled within the feature space defined by the calibration set, and guaranteeing the isomorphism of the statistical distributions of the two subsets, providing an optimal data structure basis for model training and validation.

### 2.4. Feature Selection

Numerous studies have examined the comparative accuracy of models established with either the full spectrum or specific bands following feature screening. These investigations have consistently revealed that the presence of redundant data can be significantly mitigated through the elimination of select feature bands. Such optimization not only enhances the predictive accuracy of the model but also improves its interpretability [17,37,38]. In numerous studies, Genetic Algorithms (GA) have demonstrated notable performance when applied to selected feature bands ([39]). GA possesses the capability to select the most optimal band combination by emulating the evolutionary process observed in nature [39,40]. In this investigation, GA are selected for the task of band selection in spectral data analysis. The detailed technical approach is illustrated in Figure 2. As part of the study, Lasso, Ridge, Logistic Regression (LR), Minimum Information Coefficient (MIC), and Correlation (Corr) were chosen to assess the performance of each band combination. The population size was set to 135, the chromosome length to 2050, the mutation rate to 0.03, and the crossover rate to 0.8. The evolutionary process was repeated for 100 generations. Ultimately, 22 spectral bands exhibiting distinctive characteristics were identified, primarily clustered around wavelengths of 580, 920, 1390, and 1920 nm. These bands were selected for model development. This band selection outcome aligns with the SOM-response bands identified in the near-infrared region by Ben-Dor [41]. Further studies will utilize a subset of these GA-screened characteristic spectral bands for model construction. Figure 2 provides a visual representation of the genetic algorithm’s band selection process.

### 2.5. Proposed Methodology

#### 2.5.1. Base Learners and Grid Search Optimization

In this extensive study, a comprehensive suite of 34 distinct machine learning (ML) and deep learning (DL) models were employed as the foundation learners. These models have garnered empirical evidence of superior predictive capabilities across a multitude of studies. They encompass a wide array of techniques, including linear regression models (LRM), tree-based models (TM), and neural networks (NN). LRM is good at handling linear relational data ([42]), TM Improves performance by integrating predictions from multiple trees [24]. DL uses its multi-layer NN structure to excel in several areas [43,44]. The number of TM, DL and LRM used is 9, 15 and 10, respectively. Detailed information can be found in the relevant literature [45,46,47,48,49,50,51].

In this study, several of the base learners will utilize the grid search algorithm (GS) to fine-tune their hyperparameters. The grid search algorithm is a widely used method for optimizing hyperparameters in ML models. It systematically constructs a hyperparameter space tailored to the specific nuances of each hyperparameter, organizes and intertwines a multitude of hyperparameters within the predefined range, and then meticulously selects a set of optimal hyperparameter combinations that enable the base learner to achieve its peak performance [52,53]. Table 1 list some of the hyperparameters for the base learners.

In this research, the SVR employs a linear kernel function with a penalty parameter set to 1.0; the principal components of PLSR set to three; and the Tweedie distribution’s weight exponent set to 0. The number of neighbors considered for KNR is 10, with uniform weighting. The regularization parameters alpha for Ridge, Lasso, and ENR are 1.2, and the mixing proportion for ENR is 0.65. The smoothing parameter for the GAM is 0.7. The degree of the polynomial in ARS is 9, with candidate values for the regularization strength being [0.0001, 0.001, 0.01, 0.1]. For RVR, the kernel function is ‘rbf’ with a regularization strength of 1.2, the number of components in GMMR is 3, and the covariance matrix type is ‘full’. The polynomial degree for MARS is 2, with a maximum number of spline terms set to 20; the MLP has 100 neurons in its hidden layer, uses the ‘Linear’ activation function, and employs ‘Adam’ as the optimizer. All algorithms were implemented using the scikit-learn library in a Python 3.10.0 environment.

This study utilized the TensorFlow library to construct a one-dimensional convolutional neural network (1D-CNN). The CNN comprises four convolutional layers with the number of filters increasing sequentially from 16 to 128, and filter sizes set to 3. The activation function is configured as “Linear”. Pooling layers with a window size of 2 are employed for dimensionality reduction and feature extraction. The model training process incorporates the Adam optimizer, with the mean squared error (MSE) serving as the loss function. Furthermore, LSTM, ELM, AE, GAN, and RNN regression prediction models were realized using the Keras library. Both LSTM and RNN models consist of 60 and 64 hidden units, respectively, with “Linear” activation functions, Adam optimizers, and MSE loss functions, along with a dropout rate of 0.2. The ELM model encompasses a hidden layer with 64 neurons and an output layer with a single neuron. Weights are randomly initialized, and the activation function is set to “Linear”. The generator input dimension for GAN is 24, with the hidden layer containing 256 neurons. The RNN utilizes a “Linear” activation function. The discriminator has an input of 1 and a hidden layer with 256 neurons, also employing a “Linear” activation function. The AE is constructed in Keras, where the encoder compresses the input and the decoder reconstructs the features, with the ELM structural parameters consistent with those previously described.

#### 2.5.2. Weighted Averaging

Figure 3 demonstrates the WA technology roadmap. A pivotal factor influencing WA is the strategic allocation of base learner weights. In this study, we compute the weights for each base learner using eight distinct methodologies, ultimately integrating the final outcomes based on these computed weights.

#### 2.5.3. Q-Learning

Reinforcement Learning (RL) stands as one of the most effective methods for tackling Markov Decision Problems (MDPs). It excels in capturing valuable insights from the learning trajectories generated during algorithm execution, which in turn enables the optimization of tasks [27,54]. Q-learning is a RL algorithm that assigns each base learner as an action when calculating the weights for the base learners. The EM’s weights are treated as a state. Suppose there are *K* base learners, the state can be represented as a *K*-dimensional vector [σ_1_, σ_2_, …, σ_k_], where each *σᵢ* denotes the weight of the itch base learner. The initial weights are set to an equal value (e.g., 1/*K*). In each iteration, a base learner is chosen as the action, and the EM prediction result is computed based on the current weight vector. Subsequently, the reward value is determined based on the true label. Then, a new state is calculated, and the *Q* value is updated. Q-learning updates the *Q*-value using Formula (1) to optimize the weight assignment of each base learner in different states.(1)Q(s,a)=(1−a)×Q(s,a)+(r+γ×max(Q(ś,á)))

*Q*(s,a) represents the *Q*-value of acting a in states. a is the learning rate, r is the reward value, γ is the discount factor, ś is the new state, and á is the next action. The process of updating the *Q*-value involves continuous iteration until convergence. The implementation of Q-learning weight allocation begins by defining the current weight vector. Then, the action space is set up, consisting of *K* actions, each corresponding to “fine-tuning the weights of the i-th base learner”. Next, the action is executed and the integrated prediction is calculated based on the new weights. The reward r is set according to the improvement in prediction performance; that is, the more the integrated model’s mean square error decreases, the greater the positive reward is given. Conversely, a negative reward (penalty) is given. The Q-learning iteration process begins by initializing the *Q* table, with the state sbeing an equal-weight vector and all *Q* values set to zero. In each episode (round), based on the current state s, an action *a*ᵢ is selected using the *ε*-greedy strategy, and then the action *a*ᵢ is executed to fine-tune the weight vector, obtaining the new state ś. Finally, step 2 is repeated until the weight vector converges (with minimal changes) or the maximum number of episodes is reached. After the learning is completed, for the final state (weight vector), the stable weight corresponding to the action trajectory that is evaluated as the optimal (with the highest Q value) is selected.

#### 2.5.4. Adaptive Filter

Adaptive filters (AF) can adjust their parameters in real-time based on the characteristics of the input signal, while reducing the impact of noise [55,56]. When calculating the weights for the base learner, the AF initially sets all weights to equal values. It then iteratively computes the mean squared error (MSE) of the base learner’s loss function using the Gradient Descent Algorithm and updates the weights to gradually optimize model performance. The number of iterations is set to 100. Figure 4 illustrates the schematic diagram of the AF assigning weights. The implementation details of AF weight allocation involve the following specific steps: Firstly, there are K base learners, and the initial weight vector is set. Then, iterative optimization is carried out. In each iteration, the prediction results of the base learners are weighted and integrated using the current weight w, and the final prediction of the integrated model (EM) is obtained. The mean square error (MSE) of this prediction on the validation set (or a reserved subset of the training set) is calculated as the loss function J(w). Then, the gradient descent algorithm is used to optimize the weights. The gradient ∇J(w) of the loss function J(w) with respect to the weight vector w is calculated. The direction of the gradient indicates the direction in which the weights of each base learner should be adjusted to reduce the integration error. Additionally, to ensure the meaning of the weights, they are standardized after each update to satisfy a sum of 1 and non-negativity. Finally, the above process is repeated until the preset loss function convergence is reached. The stable weights w obtained at the end are the optimal weights assigned to each base learner.

#### 2.5.5. Adaptive Learning Methods

Adaptive learning methods (AL), as a personalized approach for adjusting learning content, can optimize the learning effects of the algorithm [28]. When calculating the weights for the base learner, AL first generates initial weights using a Gaussian distribution. It then calculates the relative performance score using the Mean Absolute Error (MAE) of the base learner’s training process through Equation (2) and updates the weights using Equation (3).(2)$$S_{i}=1-\frac{e_{i}}{MAE}$$(3)$$x\prime =\frac{x-min(x)}{\max\left(x\right)-min(x)}$$

$S_{i}$ represents the relative score of the sample, where $e_{i}$ denotes the absolute error of the ith sample, *x* is the original data, and x′ is the adjusted data. The specific steps for implementing AL weight allocation are as follows: Firstly, initialize and randomly generate a set of initial weights. Then, for each base learner, during its entire training process, calculate the average absolute error of it on all training samples. Next, for the purpose of horizontal comparison, combine the errors of all base learners into a vector. Calculate the relative performance score of each base learner. Subsequently, directly use the calculated relative performance score S_i as the new weight basis. Finally, perform minimum-maximum normalization on the obtained scores, mapping them to the [0, 1] interval. Then, normalize the values after normalization so that their sum is 1, thereby obtaining a new round of weight allocation. Finally, iterate the above process, dynamically update the weights based on the latest performance of the learners until the training is completed. The final weights used are the results of the last iteration update.

#### 2.5.6. Self-Attention Mechanism

In recent years, with the rapid development of DL and the Natural Language Processing (NLP) field, the Self-Attention Mechanism (Sam), due to its efficient parallel computing capabilities and its ability to accurately understand complex structures in data, has been widely applied in feature extraction within complex Neural Networks (NNs) [29]. When calculating the weights for the base learner, the Sam learns the importance of each feature vector based on the relationships between the input feature vectors, thereby achieving weight calculation. First, each output of the base learner is treated as a feature vector, which undergoes linear transformation to obtain a feature representation. Subsequently, the similarity score between features is calculated using the dot product operation (Equation (4)). After obtaining similar scores, they are normalized using the Softmax function Equation (5), resulting in the final weights of the base learner. Figure 5 illustrates the schematic diagram of Sam calculating weight.(4)$${score}_{ij}=a(h_{i},h_{j})=h_{i}\times h_{j}$$(5)$${weight}_{ij}=softmax({score}_{ij})=\frac{exp({score}_{ij})}{\sum exp(s{score}_{ij})}$$

#### 2.5.7. Genetic Algorithm

In the process of calculating the weight of the base learner, GA will randomly generate a group of initial individuals, each representing a set of weight vectors. These individuals are then evaluated through Weighted Assignment WA EM to determine their fitness value. This value serves as a metric for assessing the performance of the base learner. Subsequently, individuals are selected for reproduction based on their fitness value using the roulette wheel method, with the fittest individuals acting as the parents for the next generation. During the crossover phase, genetic crossover (weight vector crossover) is employed to generate offspring from the parent individuals, followed by genetic mutation (weight vector variation) to introduce random perturbations and enhance population diversity. The mutation operation involves random number replacement. Lastly, some parent individuals are replaced by freshly created offspring, forming a new generation of the population. This cycle is repeated until the predefined number of iterations, 100 in this instance, is complete. The weight vector with the highest fitness value from the final generation is then selected as the final weight vector.

#### 2.5.8. Meta-Learning

By training the meta-learner, Meta-Learning can autonomously discover the features that the EM algorithm requires to address during the training process, thereby enhancing the flexibility and generalization capabilities of the base learner’s ensemble [57,58]. In the process of determining the weight of a base learner, Meta-Learning commences by initializing uniform distribution for each base learner. Subsequently, it utilizes a Regression Tree (RT) as the meta-learner for subsequent training. Throughout this training phase, the predicted output of each base learner is input, while the actual label serves as the output. The meta-learner is crafted to deliver the most advantageous weight assignments, tailored to the distinct prediction outcomes yielded by each base learner at every instance.

#### 2.5.9. Adaptive Moment Estimation

Adaptive Moment Estimation (Adam) is an optimization algorithm that combines the benefits of Momentum and RMSprop. It adjusts the learning rate adaptively to update model parameters and is renowned for its rapid convergence and robust generalization capabilities, making it a choice for optimizing NN [59]. In the study, when Adam was used as the weight calculation method, the learning rate and momentum parameters were set to 0.2, 0.7, and 0.97, respectively. For each model’s predicted and true outcomes, the loss value was computed using a predefined loss function, and the gradients of the weights with respect to the loss were obtained through backpropagation. The weights of the base learner were then updated based on these calculated gradients. Notably, Adam adaptively adjusts the learning rate for each parameter by considering both the first-order moment estimate (momentum) and the second-order moment estimate (variance) of the current gradients, leading to the update of weights. The number of iterations was set to 120.

#### 2.5.10. R^2^ Normalization

The “R^2^ Normalization” method is a commonly used technique for determining weights of the base learner in the EM. This approach involves normalizing the R^2^ scores of individual models so that their sum equals one, thus establishing their respective weights within the WA framework [60]. Additionally, “R^2^ Normalization” stands as one of the most widely recognized methods for calculating the weights of base learners in the EM algorithm. Both the scikit-learn library in Python and the Weka (Waikato Environment for Knowledge Analysis) toolkit in Java utilize this method to implement WA EM.

### 2.6. Evaluation State and Dynamic Automatic Weighting Design

To address the challenge that static evaluation indices fail to adapt to shifts in data distribution, we introduce a dynamic weight allocation method designed to enhance the performance of the EM through adaptive adjustment of the weights assigned to each base learner. This innovative approach employs a carefully crafted data structure (linked list) to store and manage both the initial and updated weights, thereby assuring the robustness and adaptability of EM to diverse datasets and application contexts.

To more accurately assess the performance of the EM algorithm, this study employed two universally acknowledged evaluation metrics: the Root Mean Square Error (RMSE) and the Coefficient of Determination (R^2^). These two measures afford a comprehensive perspective on the model’s predictive precision and its goodness-of-fit from diverse perspectives [61].

## 3. Results

### 3.1. Eight Methods to Calculate the Weights

In the current investigation, a diverse array of eight distinct base learner weight allocation methodologies was implemented, encompassing techniques such as Q-learning, self-attention mechanism (Sam), and Meta-Learning. The resultant weights attributed to the base learners via these eight methodologies are visually depicted in Figure 6. A conspicuous trend emerges: all eight approaches accord elevated weights to base learners operating within the domains of DL and Tree Model (TM), concurrently relegating lesser weights to the more traditional Regularized Linear Models (RLM). This pattern underscores the conclusion that the interrelations among features within the dataset at hand are not characterized by straightforward linearity. Evidently, RLM falls short in its efficacy and precision when compared to DL and TM in the task of assimilating and elucidating information embedded in non-linear, relational data within our contemporary, complex environment. An interesting observation pertains to the Extreme Learning Machine (ELM), which is bestowed with the most substantial weight among the trio of weight allocation strategies derived from Sam, Al, and Q-learning. This attribution suggests that ELM excels in the realm of data learning and capture. Furthermore, it is noted that models conceived under a unified conceptual framework exhibit a remarkable degree of congruity in their weight allocations, further solidifying the observed trends.

### 3.2. Performance of Ensemble Models with Different Weights in Validation Sets

To delve into the impact of varying weight allocation strategies on the efficacy of WA EM, this study harnessed seven distinct methods for computing weights that encapsulate insights gleaned from the training phase and juxtaposed these with a conventional method. The outcomes are illustrated in Figure 7. Notably, the Coefficient of Determination (R^2^) for WA EM, when applied with eight different weight distribution methodologies, achieves a score surpassing 0.85, signifying a robust predictive performance across the board. Beyond the AF variant, the performance metrics of the other six incarnations of WA EM that leverage training process insights for weight computation markedly surpass those of traditional methods. Specifically, these enhancements are reflected in a 0.004 to 0.06 increase in R^2^ and a 0.028 g kg^−1^ to 1.279 g kg^−1^ reduction in RMSE. Of particular interest, the WA EMs that assign weights through GA and Q-Learning exhibit the lowest RMSE values on the validation set, registering at 1.228 g kg^−1^ and 1.465 g kg^−1^, respectively. Additionally, the R^2^ values for WA EMs employing Sam and Q-Learning for weight assignment climb above 0.9. With the model that utilizes Sam for weight assignment achieving the highest performance on the validation set, with an impressive R^2^ of 0.927. The forthcoming research endeavors will implement the weight allocation of EM using the Sam method. The findings from this current study underscore that methodologies designed to extract information from the training process for weight computation outperform traditional methods, which rely on evaluation indicators for the same purpose. Concurrently, these results also affirm the validity and reliability of the approach that leverages insights from the model training process to determine weights.

### 3.3. Impact of Base Learners Count on Ensemble Model Accuracy

To ascertain the optimal number of WA EM based learners, this study systematically arranged the count of WA EM learners from 2 to 34, ranked by their R^2^ values in descending order, meticulously documenting the evolution of WA EM R^2^ across this range. The findings are elucidated in Figure 8. It was observed that the general efficacy of the EM ascends in tandem with the augmentation of base learners. A marked escalation in performance was conspicuous when the base learner count surged from 2 to 13, manifesting as a steep incline on the depicted curve. Within this span, the WA model’s performance underwent substantial refinement. As the base learner tally advanced to 19, the pace of enhancement decelerated yet retained a progressive trajectory. Upon reaching 26 base learners, the R^2^ metric approached a plateau around 0.927, underscoring that incremental additions of base learners thereafter yield marginal benefits to EM’s precision. Consequently, future EM investigations will adopt 26 base learners, positioning it as a strategic equilibrium between attaining superior accuracy and managing computational intricacies.

### 3.4. Computation Power Change in Ensemble Mode Under Different Weight Allocation Methods

This study conducts a comparative analysis of computational power consumption across eight distinct weight distribution methods, with the outcomes graphically represented in Figure 9. Notably, Adam demonstrates the most efficient time complexity, while the normalization technique incurs the minimal computational space requirements. Conversely, GA exhibits the highest time and space complexity, surpassing the most efficient method by 351 s and 620,040 bytes. It is pertinent to mention that earlier in the paper, we evaluated the impact of these eight weight allocation strategies on the prediction performance of the EM. The findings indicated that the EM leveraging the Sam method for weight computation achieved the most favorable results. In contrast to other methods, including GA, AF, and Adam, the Sam method imposes less stringent demands on time and space complexity. This suggests that the Sam approach not only excels in performance but also minimizes resource expenditure. Consequently, the Sam emerges as a superior choice, offering significant advantages in both EM performance and resource utilization.

### 3.5. Research on Ensemble Model Loss Under Eight Weight Distribution Modes

In the domain of ensemble learning, the ensemble of multiple base learners can sometimes lead to EM overfitting the noise and specific details within the calibration data. This overfitting results in diminished generalization capabilities on new data, a phenomenon also referred to as the overfitting of EM [21]. To explore the generalization performance of WA under eight weight allocation methods, this study conducted an in-depth analysis on EM under these eight distinct weight allocation strategies using the same loss function (MAE). Simultaneously, we implemented the shout stop mechanism, and the maximum number of iterations is configured to 160. Employing 10-fold cross-validation on the calibration set, the data for the calibration set in the graph represents the average values after 10 modeling iterations, as depicted in Figure 10. The WA EM with weights allocated by the Sam method, demonstrated the best fitting results. The WA with Sam-calculated weights was able to identify the minimum loss function in the shortest time while exhibiting the least fluctuation in the loss function. This outcome confirms that Sam enables the WA EM to possess robust generalization capabilities. It is noteworthy that the WA EM with weights assigned by the Adm and AF methods exhibited slight overfitting tendencies.

## 4. Discussion

### 4.1. Dynamic Weight Allocation

In practice, it is common to encounter situations where data distribution evolves over time. As data distribution shifts, the existing weights may become inadequate for the new data environment. This necessitates dynamic adjustments to the weights to align with the evolving data characteristics [21]. However, the traditional static weight allocation method struggles to fulfill the requirement for dynamic adjustment, thereby significantly constraining the performance of the Weighted Averaging ensemble model (WA EM) [20]. Different base learners exhibit varying strengths across distinct data subsets. Dynamic weight allocation capitalizes on this diversity, enabling ensemble models (EM) to self-adjust and thereby enhancing its generalization capabilities, making it more resilient to complex data. Simultaneously, dynamic weight adjustment mitigates the impact of interference factors on EM and reduces its sensitivity to outliers and noise. Additionally, dynamic weight allocation allows EM to better adapt to changes in data distribution. Whether dealing with intricate or nonlinear problems, or encountering variations in time and space, it can sustain the efficacy of EM through adaptive weight adjustments. It is noteworthy that in certain scenarios, dynamic weights can minimize unnecessary computational resource consumption. This is because dynamic weights enable EM to concentrate on the most critical features or aspects of the data, swiftly adapt based on the most recent data feedback, and consequently make informed decisions.

In summary, dynamic weight allocation offers a flexible mechanism that allows EM to adapt to environmental changes and data characteristics, leading to enhanced performance and adaptability across a diverse range of application scenarios.

### 4.2. The Influence of Different Weight Allocation Methods on EM

Reasonable allocation of base learner weights not only mitigates the incidence of low-value overestimation and high-value underestimation in EM predictions but also enhances the generalization performance and robustness of EM [18,19]. However, due to the diverse principles underlying them, weight allocation methods exhibit varying degrees of adaptability, each with its own merits and demerits. Currently, no single weight allocation method has proven universally suitable across all application scenarios [22,23,24]. Consequently, it is essential to conduct a systematic comparison of how weight allocation methods, influenced by their distinct principles, impact EM performance within the same application scenario. In our investigation, we employed seven distinct methods to extract pertinent information during the model training process, thereby assigning appropriate weights to the base learner. The outcomes reveal that the performance of WA EM, grounded in this theoretical framework, surpasses that of WA EM utilizing traditional evaluation indices. Employing the self-attention mechanism (Sam) method for weighing EM not only endows EM with the highest prediction accuracy but also guarantees exceptional generalization performance and robustness. Regarding this research outcome, we hypothesize that the reason may lie in Sam’s ability to compute weights by capturing interdependence among elements within sequence data during model training and translating these feature relationships into metrics of prediction accuracy and confidence scores. This speculation mirrors the operational mechanism of Sam when applied in the Transformer model [29].

To investigate the computational resources expended by the eight methods in calculating weights, this study examined two indices—time and space complexity. The findings indicate that the weight calculation method is relying on evaluation indices is comparatively straightforward and incurs lower time and space complexity during the computation of base learner weights. This outcome aligns with the findings of Dietterich et al. [62]. However, methods such as GA and QL, which extract information from the training process, demand substantial time and space resources. This is likely due to the iterative nature of GA, QL, and similar methods, which necessitate repeated cycles to identify the optimal solution when computing base learner weights, coupled with their involvement in intricate simulation behaviors and environmental interactions. Additionally, as the number of base learners grows and data complexity escalates, these methods must also perform tasks like group maintenance, crossover mutation, and other operations, leading to significant storage requirements and iteration durations [63,64,65,66]. It is noteworthy that Sam distinguishes itself among various methods that extract information from the training process to compute weights. This is due to its ability to dynamically discern relationships between inputs and adaptively learn the significance of each location, all while sustaining model performance with a minimal commitment of time and space complexity [67,68].

Overall, this study has experimentally uncovered the exceptional performance of the weight allocation method implemented with Sam in WA. The distinct advantage of Sam lies in its capacity to compute weights based on the inherent relationships within the input data, a characteristic that significantly enhances the precision of weight assignment and the expressive power of the model. Furthermore, Sam’s adaptability not only boosts the model’s capability to identify long-range dependencies but also elevates the model’s generalization performance, allowing it to more effectively adapt to diverse data distributions and task demands. Additionally, by mitigating the adverse effects of anomalous samples on the model, Sam enhances the model’s robustness.

### 4.3. Effect of Number of Base Learners on WA EM

Choosing the appropriate number of base learners stands as a pivotal determinant in the efficacy of WA EM. A plethora of research has substantiated that varying the quantity of base learners can lead to notably distinct outcomes in the overall performance of EM [69,70,71]. This aligns with the results obtained in our current research. Nevertheless, prior investigations have not delved into determining the ideal count of WAEM-based learners. Distinct from earlier research, the present study addresses the scientific inquiry of identifying the most effective number of WAEM-based learners. The findings reveal that the peak integration efficiency is achieved with a selection of 26 base learners. It is noteworthy that during our examination of how the number of base learners influences WA EM, we noticed an initial rapid ascent in the R^2^ value of EM as the count of base learners expanded. However, as additional learners were incorporated, the growth in the R^2^ value progressively decelerated. By the time the number of base learners reached 26, the R^2^ value started to stabilize at 0.927. Given this outcome, we posit that EM could become excessively tailored to the calibration data with an escalation in the number of base learners, leading to a reduction in its ability to generalize to fresh data. Concurrently, an overabundance of base learners can heighten the model’s complexity and adversely affect its interpretability. The efficacy of EM frequently hinges on the variety present among the base learners [21]. With the escalation in the number of base learners, a point of saturation is encountered where the incremental contribution of extra base learners to enhance diversity diminishes. Consequently, the upward trend in the R^2^ value decelerates progressively until it ceases to increase.

### 4.4. The Computing Power Consumption Under This Plan

In this study, we focused on the model training process and selected various methods to calculate the weights of the base learners. The results indicated that allocating weights based on the self-attention mechanism (Sam) from the perspective of the training process can make the ensemble model have the strongest performance [29]. At the same time, it can also optimally save computing power consumption. Regarding this result, we hypothesize that the reason for reducing the overall computing power consumption lies in the fact that it optimizes the traditional “post-training evaluation and then integration” two-stage lengthy process into a single-stage efficient process of “simultaneous optimization during training”, thereby achieving dual optimization in terms of time complexity and space complexity [29,72].

From the perspective of time complexity, the traditional weight allocation methods (such as those based on validation set performance) have significant computational redundancy. Firstly, k base learners need to be trained independently, with a time complexity of O (k * T_base), where T_base is the training time of a single model. Subsequently, all base learners need to be run to predict on the validation set to calculate performance metrics, which is equivalent to performing k forward propagations again, with a time complexity of O (k * N_v) (N_v is the size of the validation set). Finally, when integrating predictions based on static weights, all k models still need to be run for forward propagation, with a time complexity of O (k * N_t) (N_t is the size of the test set). The entire process is serial and repetitive. However, the dynamic weight allocation strategy based on Sam embeds the weight calculation depth into the training process. The self-attention mechanism dynamically calculates the weight contribution of each base learner based on the intermediate outputs or gradients on the same training batch. This means that training and weight calculation are synchronized, eliminating the need for an independent and additional validation evaluation stage and the O (k * N_v) evaluation overhead [72,73]. Secondly, the merging of forward propagation is also an important factor. In the integrated prediction, although the outputs of k models are still required, since the weights are dynamically generated based on the training process, the model itself may “implicitly” filter or condense the information through this process.

From the perspective of space complexity analysis, the traditional method requires simultaneously storing k completely trained base learner models and usually retains the prediction results of the entire validation set in memory for calculation. The space complexity is O (k * M_base + k * N_v), where M_base is the memory occupation of a single model’s parameters. Although the Sam method requires maintaining a small parameter matrix of the self-attention mechanism during training (its scale is only linearly related to the number of base learners k, and is usually much smaller than the base learners themselves), its core advantage lies in its “dynamicity” and “processuality”. Since the weights are functions of the training process rather than depending on a fixed, large-scale validation set result, there is no need to store intermediate predictions, which eliminates the huge memory overhead of storing the complete prediction results of all base learners on the large validation set O (k * N_v). Secondly, it promotes model simplification [29,73]. The dynamic weight mechanism can more intelligently identify and rely on key base learners, which provides clear guidance for subsequent model compression or selective integration. In practice, only the core learners with high weights and their dynamic weight generation rules can be retained, thereby significantly reducing the storage space occupation of the final deployed model O (k’ * M_base + M_sam), where M_sam is the small parameter space of the self-attention module.

In summary, the Sam strategy eliminates redundant evaluation stages by converting the weight calculation from an independent post-processing step into an embedded training process component. This reduces data repetition traversal and intermediate result storage, thereby achieving overall better utilization efficiency of computing power and memory.

### 4.5. Limitation and Future Work

The dynamic weight allocation strategy based on the self-attention mechanism (Sam) proposed in this study has demonstrated significant advantages in improving the prediction accuracy of the integrated model for SOM, but it still has several limitations and points out the direction for future research. Firstly, the model’s interpretability needs to be strengthened. The Sam mechanism can effectively capture the complex dependencies among base learners and allocate weights, but the “decision basis” represented by the attention weight matrix within it is still a black box for soil scientists [29]. The weight allocation is more based on data-driven statistical optimization rather than explicit soil science mechanisms (such as the association between specific spectral features and organic matter components), which to some extent limits the model’s conclusions’ feedback and insight into the intrinsic processes of soil science [16]. Secondly, there is still room for optimization in computational efficiency. Although dynamic weight allocation reduces the post-evaluation overhead in terms of process, the construction of a complete framework including 26 base learners and integrating the Sam mechanism requires considerable initial computational and memory resources during the training stage, which poses a practical challenge when deploying the model for large-scale soil spectral libraries or real-time monitoring applications [12]. Based on these deficiencies, future research should attempt to combine explanatory tools such as Shapley values to analyze the association between the high weights allocated by Sam and specific base learners (corresponding to specific spectral preprocessing or algorithm assumptions), and then link these findings with the known spectral responses of soil organic matter chemical components to construct a “data-driven—mechanism cognition” loop. Secondly, optimize the model architecture for efficient deployment, develop lightweight integration strategies or explore knowledge distillation techniques to compress large integrated models into a single high-performance network, thereby maintaining accuracy while significantly reducing computational and storage costs and promoting the technology towards practical business applications. Furthermore, the model constructed in this study is based on samples from a single geographical climate zone, and its superior predictive performance is verified in this homogeneous environment. The spatial heterogeneity of soil properties means that directly applying the current model to regions with significantly different soil formation environments may pose challenges [74]. One of the core future directions is to combine the dynamic weight allocation mechanism verified in this study with frameworks such as transfer learning and domain adaptation [74]. The goal is to utilize the knowledge of a large-scale and heterogeneous spectral library, through our proposed dynamic integration strategy, for rapid adaptation and refinement, thereby developing a new generation of soil spectral prediction models that maintain high accuracy in local predictions and also possess robustness across regions.

## 5. Conclusions

In this study, we introduce a suite of dynamic weight allocation strategies for base learners within an ensemble model and scrutinize the impact of varying weight computation methodologies on the performance of the Weighted Averaging ensemble model (WA EM). The principal conclusions can be summarized as follows: (1) the method of weight calculation derived from insights gleaned during the training phase outperforms traditional methods reliant on model evaluation indices; (2) the self-attention mechanism (Sam) emerges as the superior weight calculation technique, with the WA EM utilizing this method achieving an R^2^ of 0.927 and an RMSE of 2.325 g kg^−1^; and (3) the peak integration efficiency for WA EM is attained with 26 base learners. This research substantiates the efficacy of combining WAEM with VIS-NIR for SOM prediction, employing Sam’s dynamic weight allocation, thereby furnishing scientific theoretical backing for high-accuracy SOM monitoring and application of infrared technology.

The core contribution and innovation of this study lie in the proposal of a new dynamic weight allocation strategy for the weighted average ensemble model (WA EM), and its superiority has been thoroughly verified. Different from the traditional weight allocation methods that rely on static evaluation indicators, this study innovatively introduces and demonstrates the significant efficacy of the self-attention mechanism (Sam) based on learning during the training stage as the weight calculation method. This method can adaptively capture the deep dependency relationships among the prediction results of each base learner, thereby integrating their advantages more precisely. Finally, the WA EM using Sam achieved the best performance in SOM prediction and determined the optimal ensemble size of approximately 26 base learners. This provides a novel and efficient methodological framework for achieving high-precision and high-stability regional soil spectral prediction models, and promotes the practical application process of Vis-NIR spectroscopy technology in precision agriculture and soil digital mapping.

## Figures and Tables

**Figure 1 sensors-26-00195-f001:**
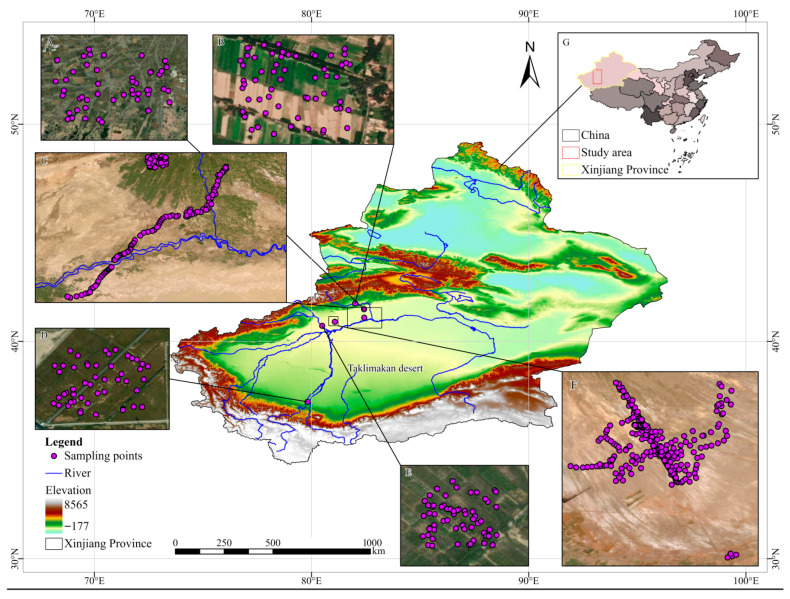
The general situation of the study area and the distribution map of the sample points in this paper (**A**): Sample distribution map of Xinhe County; (**B**): Sample distribution map of Baicheng County; (**C**): Sample distribution map of Shaya County; (**D**): Sample distribution map of Hetian County; (**E**): Sample distribution map of Avati County; (**F**): Sample distribution map of Aral County; (**G**): Wensu county.

**Figure 2 sensors-26-00195-f002:**
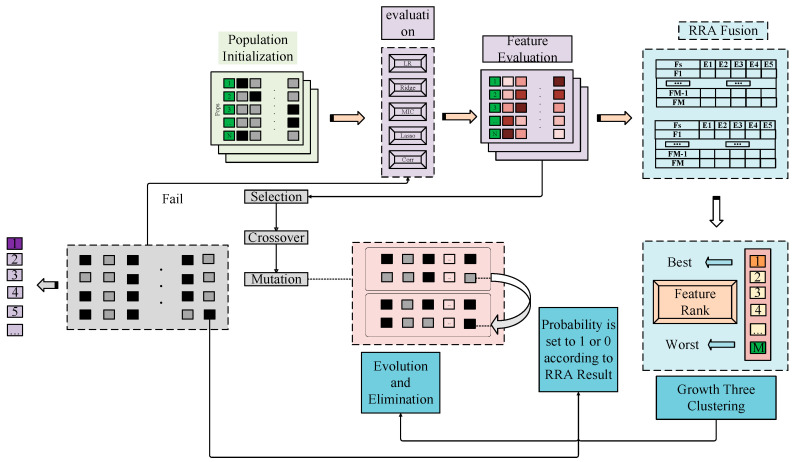
The technical roadmap of feature band screening based on genetic algorithm.

**Figure 3 sensors-26-00195-f003:**
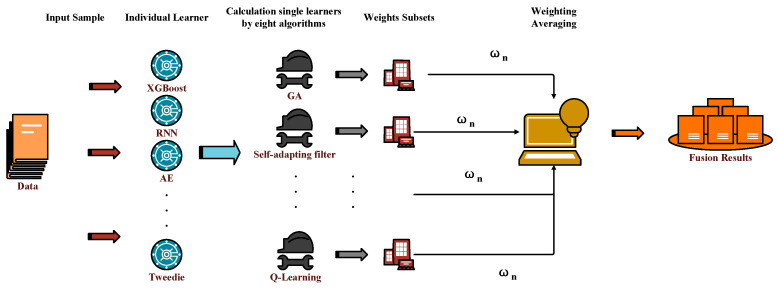
The Structure of Weighted Averaging.

**Figure 4 sensors-26-00195-f004:**
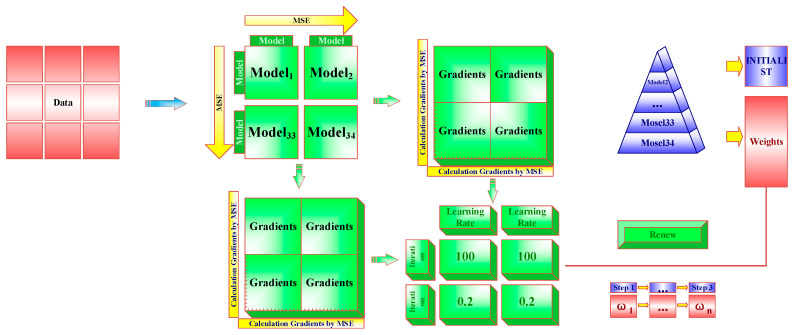
The Structure of Adaptive filter.

**Figure 5 sensors-26-00195-f005:**
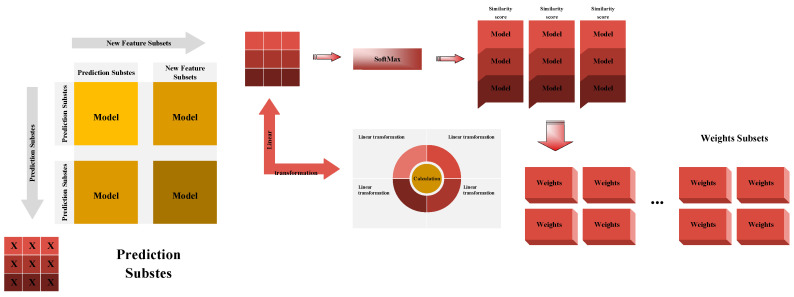
The Structure of self-attention mechanism.

**Figure 6 sensors-26-00195-f006:**
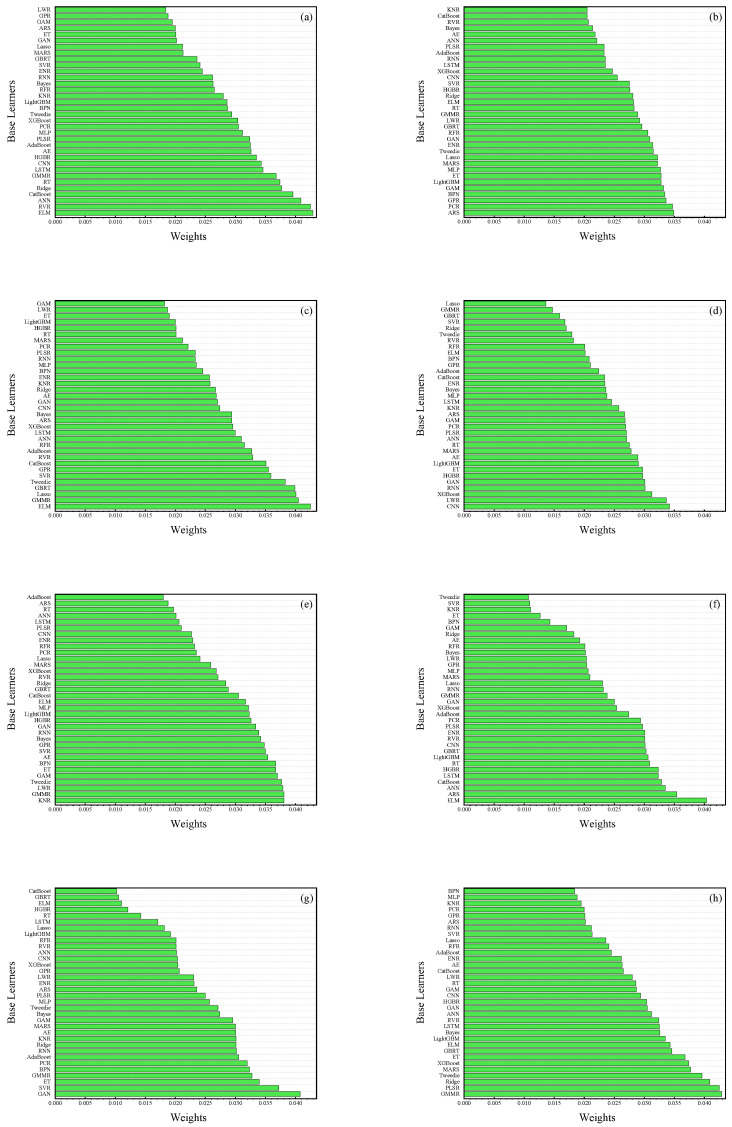
The Weights of thirty-four base learners by different calculation methods. (**a**): Q-Learning; (**b**): Adam; (**c**): Adaptive Learning; (**d**): Adaptive Filter; (**e**): Genetic Algorithm; (**f**): Self attention mechanism; (**g**): Meta Learning; (**h**): R^2^ normalization.

**Figure 7 sensors-26-00195-f007:**
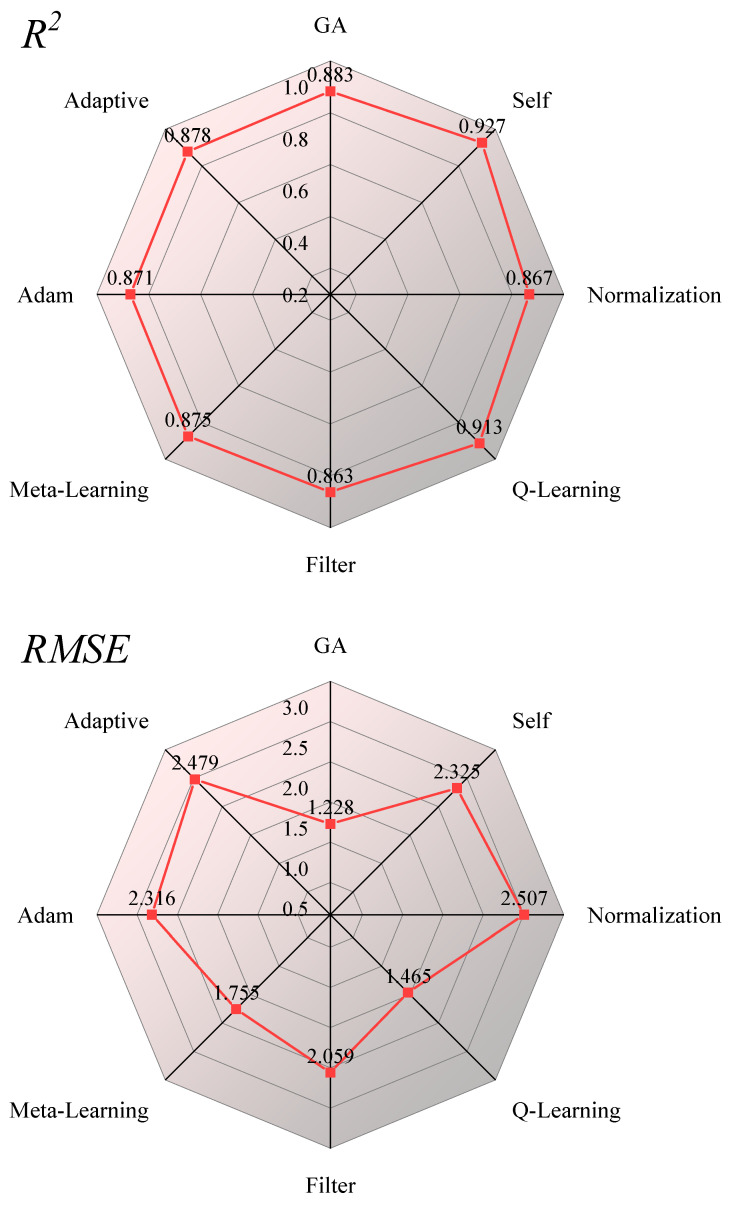
Radar map of the performance of the ensemble models in the validation set under 8 different weights.

**Figure 8 sensors-26-00195-f008:**
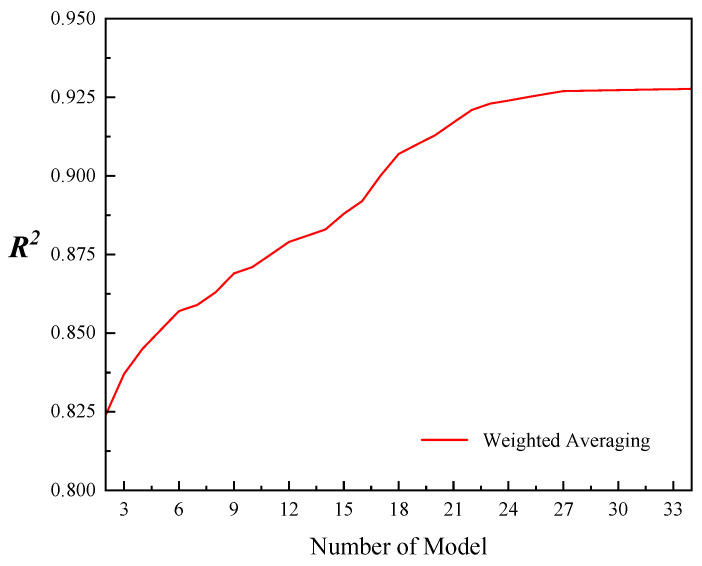
Influence of different number of base learners on ensemble models based on self-attention mechanism weighted average method.

**Figure 9 sensors-26-00195-f009:**
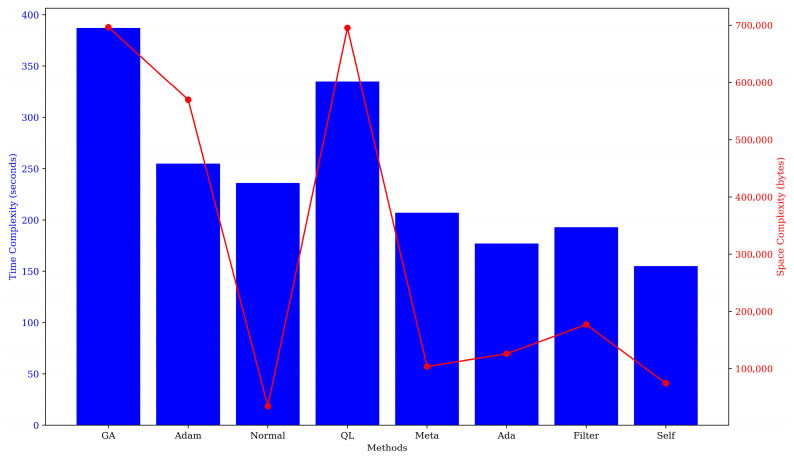
The time and space complexity of weights of 26 base learners calculated by different calculation methods.

**Figure 10 sensors-26-00195-f010:**
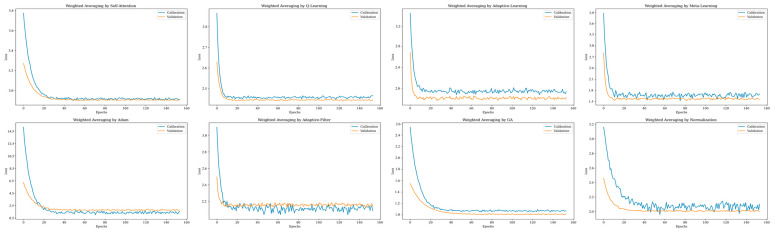
By using different calculation methods, we obtain 26 base learner weights corresponding to the ensemble models loss function images in the calibration set and the validation set.

**Table 1 sensors-26-00195-t001:** Partial model hyperparameter table.

Algorithm	Hyper-Parameter					
	Estimator	Criterion	Max_Depth	Min_Samples_Split	Min_Samples_Leaf	Learning_Rate
RT	\	mse	4	2	2	\
RF	770	mse	2	2	3	\
GBRT	560	mse	4	2	2	0.1
AdaBoost	480	mse	4	\	\	0.1
XGBoost	190	mse	3	\	\	0.2
LightGBM	130	mse	4	\	\	0.05
Extra Trees	330	mse	5	3	2	0.1
HGBR	400	mae	5	2	2	0.2
catboost	190	mse	3	\	\	0.1

## Data Availability

The data that supports the findings of this study are available on request from the corresponding author, upon reasonable request.

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
