# Peer review of "A Novel Self-Attention Mechanism-Based Dynamic Ensemble Model for Soil Hyperspectral Prediction"

_sensors, 2025, doi:10.3390/s26010195_

Round 1
Reviewer 1 Report
Comments and Suggestions for Authors
1、The description of the data processing procedure is relatively brief, lacking details on the sample handling methods. The sample partitioning strategy should be described in greater detail.
2、In the introduction, although the importance of soil organic matter (SOM) for soil health and productivity is mentioned, the discussion does not sufficiently address how SOM varies under different environmental conditions and how such variations may affect the prediction model.
3、The experimental section only compares weighted-average (WA) models with different weighting approaches, without including other ensemble strategies. This only demonstrates superiority over traditional weighting methods. It is recommended that the authors add comparisons with other classical ensemble methods to further examine whether Sam-WA outperforms alternative ensemble strategies.
4、The model is tested solely on the self-collected dataset, lacking validation of generalization ability on external datasets. The authors are advised to include soil samples collected from different years or regions, and evaluate the model’s stability across different soil types or climatic conditions.
5、The study lacks comparative validation of feature selection methods. Only the genetic algorithm (GA) is used, without comparison to other commonly used approaches (e.g., Random Forest–based methods). It is recommended that at least one additional feature selection method be introduced to assess how different approaches influence model performance.
6、Figure 10 is unclear; it is recommended that the authors revise it for improved clarity.
7、In Sections 2.5.3–2.5.6, the descriptions of Q-learning, adaptive filters, and adaptive learning methods include formulas and algorithmic steps, but the theoretical foundations and underlying principles of these methods are not explained sufficiently.
8、In Section 5 (Conclusion), although the main findings are summarized, the core contributions and innovations of the study should be explicitly highlighted.
9、In the discussion section, the authors should elaborate more deeply—for example, on issues such as the high computational cost resulting from increased model complexity.
10、In the concluding section, it is suggested that the authors include perspectives for future work, such as discussing potential limitations of the current model and proposing corresponding improvements or more specific research directions.
Reviewer 2 Report
Comments and Suggestions for Authors
The manuscript presents a new approach to machine learning in part using the method self-attention mechanism-based dynamic ensemble model.
For approbation and demonstration this approach it was applied for rather big soil sampling for hyperspectral prediction thir properties. Such works devoted to the development of new approaches to machine learning are very relevant now. Generally the goal of work was reached. Mathematical parameters of quality of ensemble model's very good. The work is suitable for the Sensors. However, all the samples were obtained from one rather arid region. In this regard, there is no reason to assume that this approach will be successful for a wider and more heterogeneous sampling. I strongly recommend mentioning this problem at the end of the introduction after the formulation of the problem and (or) in conclusion with the desire to expand the geography of research.
Reviewer 3 Report
Comments and Suggestions for Authors
ABSTRACT
- Line 19: ………. established model…… what type of model ?
- Line 17 – 29 form introductory part of the abstract, this is too long. Just a sentence or two is enough. This has made the abstract to be unnecessarily too long
- After stating the importance of the study as the introduction, then clearly state the objective(s), which is missing now
- What type of model was used, not indicated/specified in the abstract
- The R-square value reported in line 41 of the abstract shows the model is poor for the prediction. This is not encouraging
INTRODUCTION
- Line In recent years, advancements in artificial intelligence (AI) technology have introduced a suite of sophisticated methodologies including the self-attention mechanism (Sam), reinforcement learning (RL), and adaptive learning methods (AL) …………. This is a statement of fact and requires citations
http://www.doi.org/10.1016/j.jclepro.2021.130274
http://www.doi.org/10.1016/j.enconman.2021.113926
- Line 149: refining a method for computing base learner weights …… this is incomplete… using what ?
- developing a dynamic weight 149 assignment technique….. to achieve what ? this is incomplete
in all, the objective should be revise and presented more clearly
materials and methods
- To achieve this, the dataset will be split into 211 calibration and validation sets using the Kennard-Stone(K-S) algorithm …………. Change will be to was
- Line 213: This division resulted in 470 calibration samples and 234 validation points ……….. Also state their percentage
- Authors should add some model evaluation performance metrics like ME, MSE, RMSE and NRMSE which are presently missing
Result and discussion
- Figure 8 is not well presented. The plot should reflect the relationship between the observed and predicted and make a line graph with an equation and r-square
- Also, the model accuracy metrics are missing in the result, while the comparison based on these are missing in the discussion, which would have enabled good conclusion to be made
Major English Editing Required
Round 2
Reviewer 1 Report
Comments and Suggestions for Authors
The authors have made revisions according to the comments and have answered all the questions.
Author Response
Dear Reviewer 1,
We sincerely appreciate your professional comments on this manuscript. Your valuable insights have been instrumental in refining the research content and enhancing the overall quality of the paper. We are deeply grateful for the time and effort you have dedicated to this study. Thank you once again, and we wish you all the best!
Sincerely,
The Authors
Reviewer 3 Report
Comments and Suggestions for Authors
All concerns have been adequately addressed and paper is ready for publication
Comments on the Quality of English LanguageMinor English Editing Required
Author Response
Dear Reviewer 3,
We sincerely appreciate your professional comments on this manuscript. Your valuable insights have been instrumental in refining the research content and enhancing the overall quality of the paper. We are deeply grateful for the time and effort you have dedicated to this study. Thank you once again, and we wish you all the best!
Sincerely,
The Authors